# Four Novel Species and Two New Records of Boletes from India

**DOI:** 10.3390/jof9070754

**Published:** 2023-07-17

**Authors:** Kanad Das, Aniket Ghosh, Dyutiparna Chakraborty, Sudeshna Datta, Ishika Bera, Ranjith Layola MR, Farheen Banu, Alfredo Vizzini, Komsit Wisitrassameewong

**Affiliations:** 1Central National Herbarium, Botanical Survey of India, Howrah 711103, India; daskanadbsi@gmail.com (K.D.); ghosh.aniket87@gmail.com (A.G.); datta_su09@yahoo.in (S.D.); rlmr66@gmail.com (R.L.M.); banufarheen85@gmail.com (F.B.); 2Eastern Regional Centre, Botanical Survey of India, Shillong 793003, India; dyuti.parna.mail@gmail.com; 3Center of Excellence in Fungal Research, Mae Fah Luang University, Chiang Rai 57100, Thailand; iamishika6@gmail.com; 4Department of Life Sciences and Systems Biology, University of Torino, 10124 Torino, Italy; 5National Biobank of Thailand, National Science and Technology Development Agency, Pathum Thani 12120, Thailand

**Keywords:** boletales, molecular phylogeny, morphology, taxonomy, uttarakhand

## Abstract

Repeated macrofungal explorations, followed by thorough examination of species through morphology and molecular phylogeny, have made it clear that European and American names of wild mushrooms were inadvertently misapplied quite often to Asian lookalikes by mycologists/taxonomists in the past. Therefore, in order to reveal this mushroom treasure, in recent years, taxonomical research on wild mushrooms has been intensified in Asian countries, including India, by undertaking a combined approach of morpho-taxonomy and multigene molecular phylogeny. Boletoid mushrooms (Boletaceae) are no exception. While working on boletoid mushrooms of the Indian Himalayas, authors recently came across six interesting species of boletoid mushrooms. In the present communication, four novel species, namely *Leccinellum binderi*, *Cyanoboletus paurianus*, *Xerocomus uttarakhandae,* and *Xerocomellus himalayanus*, are established based on morphology and molecular phylogenetic estimations. Moreover, *Cyanoboletus macroporus* and *Xerocomus fraternus* are also reported here for the first time in India.

## 1. Introduction

Boletes represent fleshy, readily decaying (putrescent) poroid macrofungi (mushrooms) in the order Boletales of Agaricomycetes (Agaricomycotina, Basidiomycota). Mainly, these mushrooms belong to four families, namely: Boletaceae, Boletinellaceae, Suillaceae, and Gyroporaceae. They are the most popular wild-edible fleshy mushrooms and are appreciated widely across the globe. As ectomycorrhizal fungi, they play a crucial role in forest ecosystems by establishing mutual associations with forest trees. Presently, boletoid mushrooms comprise over 1270 species from around the world, belonging to 108 genera [1,2,3,4,5,6,7,8,9,10]. Earlier, the systematics of these mushrooms was mainly established based on their macro- and micromorphology. But this scenario has drastically changed during the past decade, when single- to multigene molecular phylogeny was applied in combination with morphology to revise the systematics of these mushrooms. This resulted in the discovery of several novel genera and numerous new species, especially in Asian countries.

The fungi (Mycobiota) of the state of Uttarakhand and the state of Himachal Pradesh (western Himalaya, India) are exceptionally diverse in terms of ectomycorrhizal macrofungi, as evidenced in numerous relevant literatures [11,12,13,14,15]. A focused and dedicated approach to a few other groups of ectomycorrhizal mushrooms (Russulaceae, Amanitaceae) has already been undertaken [16,17,18,19,20,21,22]. But unfortunately, serious investigation, i.e., the combined approach of molecular phylogeny and morpho-taxonomy of boletoid mushrooms, has not yet been undertaken in these states. Earlier, morphology-based part work [23] that has been undertaken in the Garhwal Himalaya (only a part of Uttarakhand) was completely mishandled. Species were wrongly identified, and the names of North American/European mushrooms were misapplied to almost all the collected specimens of the Garhwal Himalaya. Moreover, molecular phylogeny, which is the backbone for classifying boletoid mushrooms, has unfortunately not been applied to these taxa of the Garhwal Himalaya. Similarly, an account of Boletes from Himachal Pradesh was published by Lakhanpal [11], where 56 species were described under 7 genera. Inadvertently, the names of North American or European species were applied to these Indian taxa. Therefore, a holistic approach of thorough exploration followed by dedicated investigation (molecular phylogeny and morpho-taxonomy) on boletoid mushrooms distributed in the west to east of the Indian Himalayas will undoubtedly reveal several novel taxa and will also resolve many taxonomic issues in years to come. 

Recently, while undertaking routine macrofungal explorations in different parts of Uttarakhand and Himachal Pradesh, the authors came across some interesting boletoid mushrooms. A thorough examination of these collections through macro- and micromorphology, followed by multigene molecular phylogenetic estimations, uncovered four new species and two new records for Indian mycobiota. *Leccinellum binderi* sp. nov., *Cyanoboletus paurianus* sp. nov., *Xerocomus uttarakhandae* sp. nov., and *Xerocomellus himalayanus* sp. nov. are described in detail. Moreover, *Cyanoboletus macroporus* (originally reported from Pakistan) and *Xerocomus fraternus* (originally reported from China) are also reported here for the first time from India. *Xerocomus* is abbreviated as ‘*X*.’ whereas *Xerocomellus* is abbreviated as ‘*Xe*.’ in this paper.

## 2. Materials and Methods

### 2.1. Macrofungal Survey and Morphological Study

Routine macrofungal surveys were undertaken in temperate and subalpine Himalayan forests in Pauri and Rudraprayag districts (temperate mixed forests) of Uttarakhand and Chamba district (temperate coniferous forests) of Himachal Pradesh in India from 2021 to 2022. Several boletoid mushrooms were collected from both states. Macromorphological characters and habitat details were recorded in fresh, young, and mature basidiomata in the field and/or in the base camp. After recording the macromorphological characters, basidiomata were placed in a field dryer for drying. Photographs of these fresh and dry basidiomata and microphotographs were taken with the aid of Canon SX 220 HS and Nikon-DS-Ri1 (dedicated to the Nikon Eclipse Ni compound microscope) cameras. Color codes and terms used are mostly from the Methuen Handbook of Color [24]. Micromorphological characters were observed with compound microscopes (Nikon Eclipse Ni-U and Olympus CX 41). Free-hand sections from dry specimens were mounted in a mixture of 5% KOH, 1% Phloxine, and 1% Congo red or in distilled water. Micromorphological drawings were prepared with a drawing tube (attached to the Olympus CX 41 microscope) at 1000×. The basidium length excludes that of the sterigmata. Basidiospore measurements were recorded in profile view from 30 basidiospores. Basidiospore measurements and length/width ratios (Q) are recorded here as minimum-mean-maximum. Herbarium codes follow Thiers (continuously updated). Field emission scanning electron microscope (FESEM) illustrations of basidiospores were obtained from dry spores (spore prints) that were directly mounted on a double-sided adhesive tape pasted on a metallic specimen stub and then scanned with a gold coating at different magnifications in high vacuum mode to observe patterns of spore ornamentation. This work was carried out with an FEI Quanta FEG 250 model installed at the S.N. Bose National Centre for Basic Sciences in Kolkata, India. 

### 2.2. Genomic DNA Extraction, PCR Amplification and Sequencing

The genomic DNA was extracted from 100 mg of dried basidioma from five samples using a modified cetyltrimethylammonium bromide (CTAB) DNA isolation protocol [25]. The DNA quality and quantity were checked by taking absorbance readings in a NanoDrop Lite UV spectrophotometer (Thermo Scientific, Waltham, MA, USA). Genomic DNA dilutions were done for highly concentrated DNA accessions up to 50 ng/µL for PCR amplification. The PCR amplification of the Internal Transcribed Spacer region (nrITS), part of the 28S ribosomal RNA (nrLSU), region between conserved domains 6 and 7 of the second largest subunit of RNA polymerase II (*rpb2*), and part of the translation elongation factor 1-α (*tef*1-α) were done using the primer pairs ITS1-F and ITS4; LR0R and LR5; b*rpb2*-6F and f*rpb2*-7cR; and ef1-983F and ef1-1567R, respectively [26,27,28,29,30]. PCR amplification was carried out in a ProFlex PCR system (Applied Biosystems, Waltham, MA, USA) programmed for an initial denaturation at 94 °C for 3 min, followed by 35 cycles of denaturation at 94 °C for 1 min, annealing at 50 °C for 30 s, and extension at 72 °C for 1 min. The final extension was kept at 72 °C for 7 min. The PCR products were purified using the QIAquick PCR Purification Kit (QIAGEN, Hilden, Germany). Both strands of the PCR products were sequenced at Eurofins Genomics India Pvt. Ltd., Bengaluru, India. The sequence quality was checked using Sequence Scanner Software ver. 1 (Applied Biosystems). Sequence alignment and the required edition of the obtained sequences were carried out using Geneious Pro ver. 5.1 [31]. All newly generated sequences in this study were submitted to GenBank. Accession numbers of species used in phylogenetic analysis (Figure 1, Figure 2 and Figure 3) are listed in Table 1, Table 2 and Table 3.

### 2.3. Phylogenetic Analysis

The newly generated nrITS, nrLSU, *rpb2,* and *tef*1-α sequences of *Leccinellum binderi*, *Cyanoboletus paurianus*, *C. macroporus*, *Xerocomus fraternus*, *X. uttarakhandae,* and *Xerocomellus himalayanus,* plus similar ones, were retrieved from a nBLAST search against GenBank (https://www.ncbi.nlm.nih.gov/genbank, accessed on 9 May 2023), UNITE database (https://unite.ut.ee, accessed on 9 May 2023) and relevant published phylogenies [2,5,32,33,34,35,36,37]. Four datasets (nrITS, nrLSU, *rpb2,* and *tef*1-α) were created separately. All the datasets were aligned separately using the online version of the multiple sequence alignment program MAFFT v. 7 (https://mafft.cbrc.jp/alignment/software/, accessed on 21 June 2023) with the L-INS-I strategy [38]. The alignment was checked and trimmed manually with MEGA v. 7 [39]. To eliminate ambiguously aligned positions in the alignment as objectively as possible, the online program Gblocks 0.91b [40] was used. The program was run with settings allowing for smaller blocks, gaps within these blocks, and less strict flanking positions. Species delimitation was first examined using single-locus phylogenies. When significant conflict was not observed among the single-locus phylogenies, we concatenated them into one multi-locus dataset using BioEdit v. 7.0.9 [41]. The combined dataset was phylogenetically analyzed using maximum likelihood (ML). ML was performed using raxmlGUI 2.0 [42] with the GTRGAMMA substitution model. ML analysis was executed using the rapid bootstrap algorithm with 1000 replicates to obtain nodal support values. Maximum likelihood bootstrap (MLbs) values ≥70% are shown in the phylogenetic tree (Figure 1, Figure 2 and Figure 3).

## 3. Results

### 3.1. Phylogenetic Inferences

In the present study, multi-locus (nrLSU, *rpb*2, and *tef*1-α) phylogenetic analysis showed that the sequences obtained from *Leccinellum binderi* (voucher nos. KD 22-007 and KD 22-015) clustered with the *Leccinellum* lineage; however, our specimens were recovered as distinct species within the phylogenetic tree (Figure 1). On the other hand, combined three-locus (nrITS, nrLSU and *rpb*2) phylogenetic analysis revealed that the two collections of our second species, *Cyanoboletus paurianus* (voucher nos. KD 22-008 and KD 22-0009) clustered with an unidentified *Cyanoboletus* sp. (voucher nos. HKAS 90208-1 and HKAS 90208-2) from China, however, our specimens are recovered as distinct species within the phylogenetic tree (Figure 2), whereas our third species, *Cyanoboletus macroporus* (voucher nos. DC 21-02 and DC 21-24) are nested within the *C*. *macroporus* clade consisting of sample vouchers (AN-2020a and sarwar1) collected from Pakistan and suggesting its strong similarity or conspecificity with the Asian species *C*. *macroporus* with a strong (MLbs = 100%) support (Figure 2). Phylogenetic analysis based on two-locus (nrITS and nrLSU) sequences exhibits that our fourth species, *Xerocomus uttarakhandae* (voucher nos. KD 22-002 and KD 22-005), is nested (indicated in blue arrow) with an unidentified *Xerocomus* sp. (voucher nos. HKAS 90208-1 and HKAS 90208-2) with strong support (MLbs = 98%), being sister to *X*. *doodhcha* and *X*. *reticulostipitatus* (voucher nos. KD 13-082 and MEH 16_B-7) collected from India and *X*. *fulvipes* (voucher no. HKAS52556) from China; however, our specimens are recovered as distinct species within the phylogenetic tree (Figure 3). On the other hand, our fifth species, *Xerocomus fraternus* (voucher nos. KD 22-025 and KD 22-027), is nested within the *X*. *fraternus* clade consisting of Chinese collections (voucher nos. HKAS52526 and HKAS69291), suggesting its conspecificity with the Asian species, *X*. *fraternus,* with strong (MLbs = 100%) support (Figure 3). Phylogenetic analysis based on nrITS and nrLSU sequences revealed that the two collections of our sixth species, *Xerocomellus himalayanus* (voucher nos. DC 21-12 and DC 21-56) clustered with *Xerocomellus sarnarii* from Europe (voucher nos. MCVE 28571, MCVE 28577, and ML900101XE) with strong support (MLbs = 100%) and is sister to *Xerocomellus chrysenteron* collected from Europe and China; however, our specimens are recovered as distinct species within the phylogenetic tree (Figure 3).

### 3.2. Taxonomy

***Leccinellum binderi*** K. Das, A. Ghosh & Vizzini, **sp. nov.**, Figure 1, Figure 4 and Figure 5.

MycoBank: MB 848631

GenBank: OQ858380 (nrLSU, holotype), OQ858379 (nrLSU); OQ914386 (*rpb*2, holotype), OQ914387 (*rpb*2); OR102315 (*tef*1-α, holotype), OR102316 (*tef*1-α).

*Etymology*: Commemorating Dr. Manfred Binder for his significant contribution to the systematics of Boletaceae.

*Type*: INDIA, UTTARAKHAND: Pauri District, Chaubatta, 15 August 2022, alt. 1904 m, N 30°09.676′ E 78°51.240′, KD 22-007 (CAL 1923, holotype!)

*Diagnosis*: Distinguished from other known Asian species by long slender stipe, stipe context that is changing brown to black in the lower half, and the relatively large basidiospores measuring 13.8–18.22–22 × 5.4–5.96–7 μm, presence of pseudocystidia with brown to white content, occurrence under *Quercus* sp. and nrLSU, *rpb*2, and *tef*1-α sequence data. 

*Description*: Pileus 40–50 mm in diam., hemispherical to convex or planoconvex, then applanate; subtomentose, cracked (towards margin) at maturity; yellowish brown or Snuff (Brown) (5F6) or paler in combinations, greyish yellow (4B3) to champagne (4D4) towards margin; surface viscid when wet; margin entire, initially decurved then somewhat uplifted with a very narrow (up to 0.5 mm) sterile flap of tissue; turning Persian orange (6A7) with KOH but unchanging with NH_4_OH. Pore surface is pale yellow (2A3) when young, becoming orange-white (5A2) with maturity, and pale brown when bruised; pores are rounded, rarely compound, 2/mm. Tubes adnexed, 7–9 mm long, yellowish white to pale yellow (3A2–2A3), unchanging. Stipe 70–92 × 9–10 mm, more or less cylindrical with tapering apex, pithy, white to buff colored; surface subtomentose to appressed-fibrillose and scabrous; scabrous squamules white when young, becoming darker (brownish black) with maturity or handling; basal mycelia white (1A2). Context in pileus up to 8 mm thick, white to yellowish white; the context in the stipe, cream-colored, changing brown-black when exposed. The odor is mild. Spore prints were not obtained. 

Basidiospores 13.8–18.22–22 × 5.4–5.96–7 μm, [Q = 2.55–3.07–3.50], subfusoid to elongate and inequilateral in side view with distinct suprahilar depression, light yellow, smooth, inamyloid. Basidia 32–40 × 9–11.5 μm, subclavate to broadly clavate, 4-spored; sterigmata up to 6 μm long. Pleurocystidia 45–62 × 10–12.5 μm, abundant, fusoid-ventricose with subcapitate to appendiculate apex showing somewhat wrinkled in outline, thin-walled, emergent 25–30 μm. Pseudocystidia are 3–8 μm wide, abundant, filamentous, with an irregular outline, and content colorless to yellow brown. Tube edge fertile. Cheilocystidia 35–49 × 9–12 μm, abundant, fusoid-ventricose with subacute to appendiculate apex, thin-walled, emergent 9–23 μm. Hymenophoral trama boletoid, hyphae cylindrical, 4–8 μm wide. Pileipellis 100–130 μm thick, a trichoderm submerged under 15 μm thick gluten, composed of erect, frequently septate branched elements with chains of subglobose, clavate to pyriform cells; terminal cells 16–33 × 8–18 μm, cylindrical to clavate; oleiferous hyphae present. Stipitipellis is composed of a layer of slender, parallel to loosely arranged adpressed hyphae (4–10 μm wide) and frequently protruding hymenial tufts composed of basidiole, basidia, caulocystidia, and oleiferous hyphae. Caulobasidia 17–26 × 6–10 μm, clavate, 2- or 4-spored. Caulocystidia 22–30 × 9–12 μm, clavate, pyriform, ventricose to fusiform with rounded to subappendiculate or rarely mucronate apex. Clamp connections are absent.

*Habitat*: solitary or scattered, under *Quercus* sp. (Fagaceae) in temperate to subalpine Himalaya.

*Additional Specimens Examined*: INDIA, UTTARAKHAND: Pauri District, Khirsu, 17 August 2022, alt. 1774 m, N 30°10.150′ E 78°52.128′, KD 22-015 (CAL 1924); Rudraprayag District, Baniakund, 21 August 2022, alt. 2518 m, N 30°29.131′ E 79°11.653′, KD 22-032 (CAL 1925).

*Commentary*: The genus *Leccinellum* Bresinsky & Manfr. Binder was established to accommodate *Leccinum* Gray species with a yellow hymenophore and a trichodermium pileipellis [43,44,45]. Presently, this genus is represented worldwide by about 18 taxa [1,32,36,43,44,46,47,48,49,50]. *Leccinellum binderi*, the proposed new species, is characterized by the yellowish brown to snuff (brown) to greyish yellow pileus, the cream-colored stipe context that changes to brown-black on exposure, the yellowish white to pale yellow hymenophore unchanging when injured, rounded pores, long slender stipe, white unchanging stipe context, the relatively large basidiospores measuring 13.8–18.22–22 × 5.4–5.96–7 μm, and the presence of pseudocystidia with yellow-brown content. 

Now, it is realized that only with morphological features is it very difficult to separate the genus *Leccinellum* (abbreviated as *L.*) from *Leccinum* (abbreviated as *Le.*). But phylogenetically, our Indian collections fall into the genus *Leccinellum* based on a three-locus dataset (nrLSU, *tef*1-α, and *RPB*2) (Figure 1). Combining the molecular data with the morphological features of stipitate-pileate basidiomata, we place our collections in the genus *Leccinellum.*


*Leccinum parascabrum* X. Meng, Yan C. Li, and Zhu L. Yang (originally described from China) mostly exhibit the similar colors of pileus and hymenophore and the similar slender stipes with scabrous squamules; however, in *Le. Parascabrum,* the stipe context becomes greenish blue when exposed, and pileipellis shows a trichodermium nature with filamentous hyphae (never with chains of subglobose to pyriform cells) [37]. 

Two recently described species of *Leccinellum* from China, *L. alborufescens* and *L. fujianense,* are also partly related to the present species. But both *L. alborufescens* and *L. fujianense* can easily be separated from *L. binderi* by a rugulose or pitted brown to dark brown pileus, reddening of the hymenophore, and context (on bruising or exposure). Moreover, *L. alborufescens* occurs in tropical regions, whereas *L. fujianense* is found in subtropical regions [36]. 

***Cyanoboletus paurianus*** K. Das and A. Ghosh, **sp. nov.**, Figure 2, Figure 6 and Figure 7.

MycoBank: MB 848632

GenBank: OQ859919 (nrLSU, holotype); OQ859920 (nrLSU); OQ914388 (*rpb*2, holotype); OQ914389 (*rpb*2).

*Etymology*: Referring to the type locality (Pauri district) in the state of Uttarakhand (India).

*Type*: INDIA, UTTARAKHAND: Pauri District, Phedkhal, 15 August 2022, alt. 1871 m, N 30°09.579′ E 78°51.313′, KD 22-009 (CAL 1926, holotype!)

*Diagnosis*: Distinguished from other known Asian species by smaller basidiospores (9.1–11.51–13.2 × 4.3–4.85–5.5 μm), absence of gluten on the hymenial surface and basidia, and occurrence under *Quercus* sp. and nrITS, nrLSU, and *rpb*2 sequence data. 

*Description*: Pileus 40–50 mm in diam., convex when young, broadly convex with maturity (but never plane), glabrous to leathery, dark brown (8E–F4); surface viscid to sticky when wet; margin entire, incurved with a narrow flap of tissue (up to 1 mm wide); turning reddish brown (9F8) with KOH, reddish brown (9E7) with NH_4_OH, dull green (25E3) with FeSO_4_. Pore surface maize (yellow) (4A6), becoming grayish green (25E5) when bruised, then turning brownish after 2 h; pores rounded, 2/mm. Tubes adnexed, 2–2.5 mm long, maize (yellow) (4A6), becoming greyish green (25E5) when bruised. Stipe 45–55 × 7–10 mm, more or less cylindrical with slightly tapered at apex, pithy; apex (5–8 mm) concolorous to pore surface, i.e., maize (yellow) (4A6); below onwards along the length of the stipe, concolorous to pileus, i.e., reddish brown (8E–F7); glabrous to leathery, with white basal mycelia. Context in pileus up to 4–9 mm thick, cream to buff colored, changing instantly to bluish grey to greyish blue (22–23B3–5); the context in stipe buff to yellow at apex, gradually greyish black towards the base, changing bluish on exposure. Spore print is olivaceous brown. 

Basidiospores 9.1–11.51–13.2 × 4.3–4.85–5.5 μm, [Q = 1.65–2.38–2.86], subfusoid to elongate and inequilateral in side view with distinct suprahilar depression, light yellow, smooth, inamyloid. Basidia 25.3–35 × 8–9 μm, subclavate to broadly clavate, 4-spored; sterigmata up to 6 μm long. Pleurocystidia 25–35 × 5.5–7.5 μm, abundant, fusoid-ventricose with rounded to subacute apex, thin-walled, emergent 9–14 μm. The tube edge is sterile. Cheilocystidia 21–32 × 6.5–9 μm, abundant, fusoid-ventricose or rarely clavate, thin-walled, emergent 9–15 μm. Hymenophoral trama boletoid, hyphae divergent, cylindrical, glutinous, 4–5 μm wide. Pileipellis 200–300 μm thick, an ixocutis to ixotrichoderm, submerged under very thin gluten, composed of suberect to erect, frequently septate interwoven hyphae with elongate (never inflated) cells; terminal cells 32–55 × 5–8 μm, cylindrical with a rounded to subfusoid apex. Stipitipellis is composed of erect to suberect, somewhat interwoven hyphae forming trichodermium, often with tufts of abundant cystidia and some basidia. Caulocystidia 28–35 × 8–10 μm, clavate to subclavate or fusoid-ventricose, aseptate to septate with rounded to subfusoid apex. Caulobasidia 22–30 × 7–9 μm, rare, narrowly to broadly clavate, 2- or 4-spored; sterigmata up to 8 μm long. Clamp connections are absent.

*Habitat*: scattered to gregarious, under *Quercus* sp. (Fagaceae), in temperate broadleaf forest.

*Additional Specimen Examined*: INDIA, UTTARAKHAND: Pauri District, Phedkhal, 15 August 2022, alt. 1736 m, N 30°08.723′ E 78°51.212′, KD 22-008 (CAL 1927).

***Cyanoboletus macroporus*** Sarwar, Naseer & Khalid, Figure 2, Figure 8 and Figure 9.

GenBank: OQ860238 (nrITS), OQ860240 (nrITS); OQ860239 (nrLSU), OQ860241 (nrLSU); ON364552 (*rpb*2), OQ876894 (*rpb*2).

*Description*: Pileus 15–60 mm in diam., mostly convex, sometimes planoconvex with maturity, applanate with uplifted margin, reddish brown (8–9F6–8) when young to dark brown (8–9D8); surface viscid when moist, velvety, often with patches of appressed small squamules; margin entire to undulated, initially incurved then uplifted; turning greenish black (20F8) when bruised, brownish yellow (5C8) with NH_4_OH, and dark red (10C8) with KOH. Pore surface: yellowish brown (2A5), becoming blue-black (20D5) when bruised; pores: angular, pore stuffed when young; pore mouth: red (10C–D8), 0.7–0.9/mm. Tubes are adnate, 3–6 mm long, light yellow (2A5), becoming greyish blue (20D5) when bruised. Stipe 30–55 × 5–8 mm, more or less cylindrical, solid, yellow at apex, brownish black towards the base, white to buff colored at base; surface faintly pruinose. Context in pileus up to 10 mm thick, yellowish white, instantly turning bluish green (25B6) when exposed. Odour mild. Taste none. Spore print is olive brown. 

Basidiospores 11–12.5–13.4 × 4.6–5.2–6 μm, [*n* = 30, Q = 2.1–2.2–2.6], ellipsoid to fusoid and inequilateral in side view, hyaline, smooth under light microscopy. Basidia 26–34 × 7–11 μm, clavate, 4-spored; sterigmata 2–4 × 1–2 μm. Pleurocystidia 45–67 × 10–12 μm, subcylindric to fusiform or subventricose with rounded apex, thin-walled, emergent up to 35 μm. Subhymenial layer 10–20 μm thick. Tube edge fertile. Cheilocystidia were not found. Hymenophoral trama divergent, hyphae cylindrical, septate, unbranched, 2–4 μm wide. Pileipellis is 60–150 μm thick, a trichoderm composed of erect chains of cells; terminal cells are 6–15 × 4–8 μm, broadly cylindrical to subventricose. Stipitipellis fertile, composed of slender, subparallel hyphae (3–5 μm wide); sometimes protruding hymenial tufts composed of basidia and cystidia. Caulobasidia 26–35 × 6–8 μm, subcylindrical to clavate, 2- or 4-spored. Caulocystidia 29–35 × 7–9 μm, subcylindric to subfusoid. Clamp connections are absent.

*Habitat*: Solitary to gregarious, in temperate coniferous forests of *Cedrus deodara*.

*Specimens Examined*: INDIA, HIMACHAL PRADESH: Chamba District, Kalatop, 18 July 2021, alt. 2398 m, N 32°32.076′ E 76°00.931′, DC 21-02 (CAL 1934); Kalatop, 19 July 2021, alt. 2374 m, N 32°33.051′ E 76°01.138′, DC 21-24 (CAL 1935).

*Commentary*: The genus *Cyanoboletus* is distinct from all other genera of Boletaceae by its yellowish brown, brown to dark brown pileus that shows instant bluing of context on exposure and hymenophore when bruised, cutis, trichoderm pileipellis, and smooth basidiospores [32]. But based only on morphology, it is difficult to separate most of the species. In the field, *C. paurianus* is quite close to another Asian species, *C. sinopulverulentus* (also reported below from India); however, the latter one shows larger basidiospores, while our proposed new species is clearly recovered in multigene phylogeny (Figure 2) [51]. Previously, *C. hymenoglutinosus* D. Chakr., K. Das, A. Baghela, S.K. Singh, and Dentinger from India could easily be distinguished from *C. paurianus* by its highly glutinous hymenial layer and basidia, which are distinctively covered with thick gluten and larger basidiospores (12–12.8–15 × 4.8–5.2–5.8 μm) [47]. Our second species, *C. macroporus,* is a recently established (2021) species from the temperate to subalpine forests of Pakistan. This species can be distinguished by its brownish-red pileus that instantly changes its color to olivaceous black to dark greenish black when handled, much wider pores among the other similar *Cyanoboletus* species, stipe without reticulation, yellow to yellowish brown hymenophore with angular pores, and nrITS-based phylogeny. Our Indian collection shows morphological similarities and phylogenetic support to establish its conformity with the Pakistani species [52]. Moreover, in this present study, some characters that are missing in the protologue, like pore size, microchemical spot test on pileus, and context, are also recorded here. 

***Xerocomus uttarakhandae*** K. Das, Sudeshna Datta, and A. Ghosh, **sp. nov.**, Figure 3, Figure 10 and Figure 11.

MycoBank: MB 848633

GenBank: OQ748036 (nrITS, holotype); OQ748035 (nrITS); OQ748037 (nrLSU, holotype); OQ748038 (nrLSU).

*Etymology*: Referring to the type locality (the state of Uttarakhand), India.

*Type*: INDIA, UTTARAKHAND: Pauri District, Teka, 14 August 2022, alt. 1843 m, N 30°06.878′ E 78°45.485′, KD 22-005 (CAL 1928, holotype!).

*Diagnosis*: Distinguished from other closely allied Asian species by cracked to areolate pileus surface showing reddish context, shorter stipe, absence of reticulation on stipe surface, occurrence under *Quercus* sp., and nrITS and nrLSU sequence data. 

*Description*: Pileus 37–70 mm in diam., hemispherical to convex or planoconvex; subtomentose to velvety, becoming cracked to areolate at maturity; greyish orange to greyish brown (5B–D3) or paler in combinations, showing reddish areas through cracks and cut or injured areas beneath the cuticle; surface never viscid when wet; margin entire, initially decurved then somewhat uplifted with a very narrow (up to 0.5 mm) sterile flap of tissue; turning brown (6E7) with KOH and olive gray (3E2) with FeSO_4_. Pore surface: pastel yellow (2A4) or lemon yellow, becoming greyish turquoise (24D5–6) when bruised, then brownish after some time; pores: angular, often compound, 1/mm. Tubes adnexed, 6–8 mm long, pastel yellow (2A4) or lemon yellow, color reaction same as pore surface. Stipe 24–60 × 5–13 mm, more or less cylindrical, gradually tapering towards the base, longitudinally striate to fibrillose at the upper half, yellowish white at the apex, orange white (6A2) towards the middle and lower half, with white (1A2) basal mycelia. Context in pileus up to 12 mm thick, white to yellowish white; the context in the stipe, yellowish white, turning pale yellow (4A3) with KOH and greyish green (25C3) with FeSO_4_. Odour mild. Spore prints were not obtained. 

Basidiospores 9–10.6–12.6 × 3.8–4.5–5.1 μm, [*n* = 30, Q = 2–2.36–2.74], ellipsoid to fusoid and inequilateral in side view, hyaline, smooth under light microscope but under SEM spore surface bacillate. Basidia 24–32 × 8–8.5 μm, clavate, 4-spored; sterigmata 2–4 × 1–2 μm. Pleurocystidia 38–62 × 7–10 μm, subcylindric to ventricose, or subfusoid, thin-walled, few with incustrations on wall, emergent up to 39 μm. Subhymenial layer 12–15 μm thick. Tube edge fertile. Cheilocystidia 34.5–47 × 7.5–11 μm, abundant, clavate to subfusoid or fusoid with tapering apex, thin-walled, emergent 12–20 μm. Hymenophoral trama phylloporoid, hyphae cylindrical, septate, branched, thin-walled, non-gelatinous, 5–10 μm wide. Pileipellis up to 180 μm thick, as a trichodermium, composed of erect cylindrical septate hyphae; terminal cells 25–51 × 7–10 μm, cylindrical, sometimes tapered at the apex; pigmented. Stipitipellis fertile composed of a layer of slender, parallelly arranged adpressed hyphae (5–8.75 μm wide) and frequently protruding hymenial tufts composed of basidia, basidioles, and caulocystidia. Caulobasidia 24–33 × 7–11 μm, rare, clavate, 4-spored. Caulocystidia 21–33 × 7–13 μm, ventricose-fusoid, clavate to bulbous, or pyriform to subcapitate. Clamp connections are absent.

*Habitat*: solitary or scattered, in temperate forests under *Quercus* sp. (Fagaceae).

*Additional Specimen Examined*: INDIA, UTTARAKHAND: Pauri District, Teka, 14 August 2022, alt. 1893 m, N 30°06.656′ E 78°45.288′, KD 22-002 (CAL 1929).

***Xerocomus fraternus*** Xue T. Zhu and Zhu L. Yang, Figure 3, Figure 12 and Figure 13.

GenBank: OQ776920 (nrITS), OQ776919 (nrITS), OQ771932 (nrLSU), and OQ771933 (nrLSU).

*Description*: Pileus 35–75 mm in diam., mostly convex, sometimes planoconvex with maturity, subtomentose, greyish orange (5B3–5) to reddish brown; surface often warty; margin entire to undulated, initially incurved then decurved to upturned with a narrow (up to 1.5 mm) sterile flap of tissue; turning light brown or sunburn (6D5) with KOH, violet brown (11E5) with NH_4_OH, and dull green (25D3) with FeSO_4_. Pore surface: light yellow or sun yellow (2A5), becoming greyish green (25E5) when bruised; pores: angular, mostly pentagonal to irregular, rarely compound, 1–2/mm. Tubes adnexed, 7–9 mm long, light yellow or sun yellow (2A5), becoming greyish green (25E5) when bruised. Stipe 40–80 × 6–8 mm, more or less cylindrical with tapering base, solid, light yellow or sun yellow (2A5) at apex, greyish yellow (4B4) towards the base, white to buff colored; surface longitudinally fibrillose; basal mycelia white (1A2). Context in pileus up to 10 mm thick, white to yellowish white, slowly becoming greenish grey (25B2) when exposed; the context in the stipe, yellowish white to cream on the upper half but reddish brown on the lower half. Odour mild. Spore print is olive brown. 

Basidiospores 8.2–10.5–12.4 × 3.2–4.61–5.6 μm, [*n* = 30, Q = 1.73–2.29–2.7], ellipsoid to fusoid and inequilateral in side view, hyaline, smooth under light microscope, but under SEM spore surface bacillate. Basidia 24–39 × 6–12 μm, clavate, 4-spored; sterigmata 3–5 × 1–2 μm. Pleurocystidia 52–99 × 7–20 μm, subcylindric to fusiform or subventricose with rounded or rarely mucronate apex, thin-walled, emergent up to 36 μm. Subhymenial layer 12.5–20 μm thick. Tube edge fertile. Cheilocystidia 38–45 × 6–8 μm, rare, subcylindrical to subfusiform, thin-walled, emergent up to 34 μm. Hymenophoral trama divergent, hyphae cylindrical, septate, unbranched, thin-walled, non-gelatinous, 3–5 μm wide. Pileipellis is 150–200 μm thick, a trichoderm composed of erect chains of cells; terminal cells are 23–48 × 6–13 μm, cylindrical, conic, subventricose, or subclavate. Stipitipellis fertile is composed of a layer of slender, parallel hyphae (5–10 μm wide) and frequently protruding hymenial tufts composed of basidia and cystidia. Caulobasidia 26–38 × 6–7 μm, subcylindrical to clavate, 4-spored. Caulocystidia 24–44 × 6–10 μm, subcylindric to clavate. Clamp connections are absent.

*Habitat*: solitary to gregarious, under *Quercus* sp. (Fagaceae) in temperate to subalpine Himalaya.

*Specimens Examined*: INDIA, UTTARAKHAND: Rudraprayag District, Chopta, 19 August 2022, alt. 2846 m, N 30°28.995′ E 79°10.760′, KD 22-025 (CAL 1930); Baniakund, 20 August 2022, alt. 2518 m, N 30°29.131′ E 79°11.653′, KD 22-027 (CAL 1931).

*Commentary*: The genus *Xerocomus* Quél. is separated from all other genera of Boletaceae by its long tubes with relatively large pores (1–3 mm in diam.), a trichodermium pileipellis, and usually bacillately warted basidiospores under SEM. Species in this genus are also quite difficult to separate by morphology alone. Therefore, molecular analysis plays a significant role in separating the species.

*Xerocomus uttarakhandae* is characterized by medium-sized basidiomata with a velvety and cracked to areolate pileus surface showing reddish context, a yellow pore surface that becomes bluish when bruised, stipe yellowish white at the apex, orange white at the mid and lower half with white basal mycelia, the presence of variously shaped (subcylindric to clavate to bulbous to pyriform) and septate caulocystidia, and their occurrence under *Quercus* sp. in temperate Himalaya. Combining morphology and molecular phylogeny, three species in this genus were erected in India in the last decade. They are *Xerocomus doodhcha* K. Das, D. Chakr., Baghela, S.K. Singh, and Dentinger; *X. longistipitatus* K. Das, A. Parihar, D. Chakr., and Baghela; and *X. reticulostipitatus* Hembrom, D. Chakr., A. Parihar, and K. Das. All three species grow under trees belonging to Fagaceae and are partly related to the presently described *Xerocomus uttarakhandae* (considering morphology and sequence data) (Figure 3). But *Xerocomus doodhcha* can be separated in the field from *X. uttarakhandae* by possessing pileus without a cracked surface and microscopically with larger caulocystidia (20–44 × 5–11 µm) [34]. *Xerocomus longistipitatus* has a distinctively long (70–185 × 10–24 mm) stipe, an ixotrichoderm nature of pileipellis, and larger basidiospores (10.8–14.6 × 3.6–4.5 µm) [33], whereas *X. reticulostipitatus* has a distinct reticulation on the stipe surface, larger basidiospores (10.3–15.6 × 3.7–5.3 µm), and larger pleurocystidia (45–66 × 9.5–13 µm) [35]. Another species, *X. subtomentosus* (Fries) Quélet (described from Europe), is somewhat similar to *X. longistipitatus* (DC 15-056); however, *X. subtomentosus* differs from *X. uttarakhandae* in possessing distinctively longer (10.5–15.2 µm) basidiospores, an olive brown to olive yellow pileus, and a longer (40–100 mm) stipe [53,54]. Phylogenetically, *X. fulvipes* Xue T. Zhu & Zhu L. Yang (originally described from China) is also close to *X. uttarakhandae* (Figure 3). but *X. fulvipes* shows a distinctively larger pileus (30–110 mm in diam.), which never shows a cracked or areolate surface (cracked showing reddening of context in *X. uttarakhandae*), and a distinctively larger stipe (30–90 × 5–13 mm) [32].

Our second species, *Xerocomus fraternus,* is distinguished by a set of characters: a light yellow pore surface and tubes; basidiospores of length measuring ≤13 µm; the lower half of the stipe context that is mostly reddish brown on exposure; and its occurrence in temperate to tropical forests [32]. Present Indian collections completely agree morphologically with the samples (holotypes) reported from the neighboring country, China. For the first time, it is being reported from India. 

***Xerocomellus himalayanus*** D. Chakr and A. Ghosh, **sp. nov.**, Figure 3, Figure 14, Figure 15 and Figure 16.

MycoBank: MB848680

GenBank: OQ847832 (nrITS, holotype); OQ847959 (nrITS); OQ847962 (nrLSU, holotype); OQ847979 (nrLSU).

*Etymology*: Referring to the type locality (the western Himalayas), India.

*Type*: INDIA, HIMACHAL PRADESH: Chamba District, Kalatop, 19 July 2021, alt. 2374 m, N 32°33.051′ E 76°01.138′, DC 21-12 (CAL 1932, holotype!)

*Diagnosis*: Distinguished from its closely allied species by its unchanging pore surface, tube, and context on exposure, yellow subpellis, occurrence under *Cedrus* sp., and nrITS and nrLSU sequence data. 

*Description*: Pileus 23–80 mm in diam., solitary, mostly convex, sometimes planoconvex with maturity, greyish yellow (3C4–5) when young to brown (6E5–7) with maturity; surface areolate when mature, showing yellow context, turning deep yellow (4A7–8) with KOH, no reaction with NH_4_OH and FeSO_4_; margin entire, sterile flap of tissue not present. Pore surface: maize yellow to deep yellow (4A6–8), no change in bruising; pores: angular, compound, 8–10/mm. Tubes adnate, 5–8 mm long, yellowish white (2A2), no change when bruised. Stipe 40–90 × 10–30 mm, mostly cylindrical to narrowly clavate, sometimes with bulbous base, solid but mostly infested with larvae, light yellow or sun yellow (2A5) at upper one third, brownish red (8C6–7) with combination of dark brown near base, greenish blue when bruised (not instantly), then finally blackish brown; surface longitudinally fibrillose; basal mycelia white (1A2), sometimes forming rooting base. Context in pileus up to 8 mm thick, yellow, no change when exposed; the context in the stipe, yellowish white, slightly turning greenish blue with time when exposed. Odor is acidic. Spore print is olive brown. 

Basidiospores 13–15.8–17 × 5.8–6.6–7.4 μm, [*n* = 30, Q = 1.87–2.21–2.68], ellipsoid to fusoid and inequilateral in side view, often with a truncate apex, hyaline, smooth under light microscopy. Basidia 37–48 × 10–14 μm, clavate, 2 to 4-spored; sterigmata 3–4 × 1–2 μm. Pleurocystidia 45–80 × 8–13 μm, subcylindric to fusiform, ventricose with rounded, thin-walled, few are brown pigmented, emergent up to 20 μm. Subhymenial layer 12–20 μm thick. Tube edge fertile. Cheilocystidia 40–45 × 8–10 μm, rare, subventricose to subfusiform, thin-walled. Pileipellis 110–150 μm thick, a palisadoderm composed of erect brown pigmented and highly incrusted hyphae, incrustation in a ladder-like pattern; terminal cells 18–51 × 6–12 μm, cylindrical to fusoid. Stipitipellis fertile near the apex of the stipe, composed of parallel hyphae (4–7 μm wide); few protruding hymenial tufts composed of basidia and cystidia. Caulobasidia 26–38 × 6–7 μm, subcylindrical to clavate, 4-spored. Caulocystidia 33–45 × 11–15 μm, subclavate to clavate, fusoid with rounded to rarely appendiculate apex. Clamp connections are absent.

*Habitat*: solitary, in temperate forests under *Cedrus deodara* (Pinaceae). 

*Additional Specimen Examined*: INDIA, HIMACHAL PRADESH: Chamba District, Kalatop, 22 July 2021, alt. 2391 m, N 32°32.550′ E 76°01.317′, DC 21-56 (CAL 1933).

*Commentary*: The genus *Xerocomellus* Šutara is separated from its morphologic sister genus *Xerocomus* by its smooth or longitudinally striate (never bacillate) basidiospore, palisadoderm nature of pileipellis, small or mostly medium-sized, often vividly colored, surface dry, at first velvety and later often rimose-areolate, and a minutely granulose, sometimes longitudinally striate but mostly non-reticulate stipe, which is usually slender and not very firm [55]. Our newly proposed Indian collection features medium-sized basidiomata, brown pileus that turned areolate with maturity, deep yellow pores that remain unchanged when bruised, yellow pileus context, unchanging when exposed, stipe cylindrical to sometimes bulbous at base, yellow stipe with combination of brownish red and dark brown towards base, smooth basidiospore, and occurrence under *Cedrus deodara* in temperate coniferous forest of Western Himalaya. *Xerocomellus himalayanus* is morphologically as well as phylogenetically close to the European species *Xe. sarnarii* Simonini, Vizzini, and U. Eberh, but can be separated in the field as the latter shows a bluish color when context, tubes, and pores are bruised or exposed to air, and its occurrence under *Quercus* sp. Moreover, *Xe. sarnarii* shows smaller basidiospores (13.8–15.1 × 5.5–6.1 μm) and pleurocystidia (35–52 × 6–11 μm) [56]. Some morphologically similar and phylogenetically close members of this Indian species are *Xe. poederi* G. Moreno, Heykoop, Esteve-Rav., P. Alvarado, and Traba, and *Xe. chrysenteron* (Bull.) Šutara, but *Xe. poederi* differs from *Xe. himalayanus* by its reddish epicutis, depressed pores, reddish stipe context, and habitat under *Quercus* sp. [57]. Similarly, *Xe. chrysenteron* can be distinguished by its reddish cracks on pileus, context turning faint blue and then finally reddish on exposure, and narrower basidiospores (12.3–16.1 × 4.1–5.6 μm) without any truncation at apices [54,58]. *Xerocomellus mendocinensis* (Thiers) N. Siegel, C.F. Schwarz, and J.L. Frank; *Xe. dryophilus* (Thiers) N. Siegel, C.F. Schwarz, and J.L. Frank; and *Xe. diffractus* N. Siegel, C.F. Schwarz, and J.L. Frank are distinguished from the Indian collection by their geographical location, ecology, and nrITS and nrLSU sequences. Moreover, *Xe. Mendocinensis* differs by its pink scabrous stipe and instantly bluing tubes when bruised. Similarly, *Xe. Dryophilus* and *Xe. Diffractus* both show bluing of the hymenophore and context on exposure, which makes them distinct from *Xe. himalayanus* in the field [59]. 

## 4. Discussion 

India, with its luxuriant forests of coniferous and/or deciduous trees, is immensely diverse in terms of fleshy mushrooms. Boletoid mushrooms (Boletes) are no exception. To date, about 96 species belonging to 27 genera have been reported in Boletes from India [60,61,62,63,64,65,66,67,68]. Major ectomycorrhizal host trees that support the growth and development of these mushrooms belong to genera like *Quercus* L., *Castanopsis* (D. Don) Spach, *Lithocarpus* Blume, *Hopea* L., *Betula* L., *Shorea* Roxb. ex C.F. Gaertn., *Abies* Mill., *Picea* A. Dietr., *Cedrus* Trew, *Pinus* L., *Tsuga* (Endl.) Carrière, and *Larix* Mill. Keeping in view the existing number of genera of Boletes from the globe (108) and the reported number of genera in India (27), it becomes clear that this group is seriously under-explored, and multigene molecular phylogeny (the backbone of systematics in Boletoid mushrooms) has hardly been applied to reveal the diversity of this group. The present contribution with morphotaxonomy and multigene molecular phylogeny is an initiative to uncover this immensely diverse wealth of Boletoid mushrooms in the Indian Himalayas. The reporting of six species of Boletes in the present article brings the total to 102 species from this vast country. The four genera dealt with in this contribution are separated by the key given below. Following the present study, many macrofungal surveys will be conducted in this region to unveil the hidden diversity of Boletoid mushrooms in the near future.

### Key to the Studied Genera of Boletes

1Stipe surface with scabs that turns brownish black when bruised; stipe context is cream colored, changing brown-black when exposed ……………………*Leccinellum*
1a.Basidioma with different combinations of features ……………………………2

2Pileus surface is sticky; pileus surface, pore surface, stipe surface, and context turn dark blue instantly when bruised or exposed ……………………………*Cyanoboletus*2a.Pileus surface velvety; only pore surface turns blue slowly when bruised …3

3Pileipellis as a trichodermium; spores with bacillate ornamentation (under SEM) …….………………………………………………………………………*Xerocomus*3a.Pileipellis as a palisadoderm; spores smooth ………………………*Xerocomellus*.

## Figures and Tables

**Figure 1 jof-09-00754-f001:**
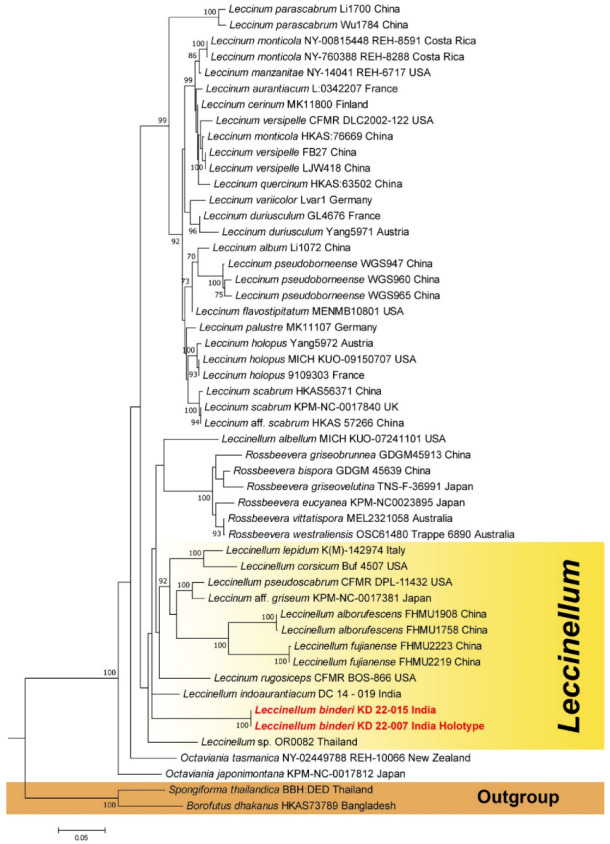
Phylogram generated by Maximum Likelihood analysis based on combined sequence data of nrLSU, *RPB*2, and *tef*1-α for *Leccinellum binderi* and allied species. Maximum likelihood bootstrap support values (MLbs) ≥ 70% are shown above or below the branches at nodes. *Leccinellum binderi* is placed in bold red font to highlight its phylogenetic position in the tree.

**Figure 2 jof-09-00754-f002:**
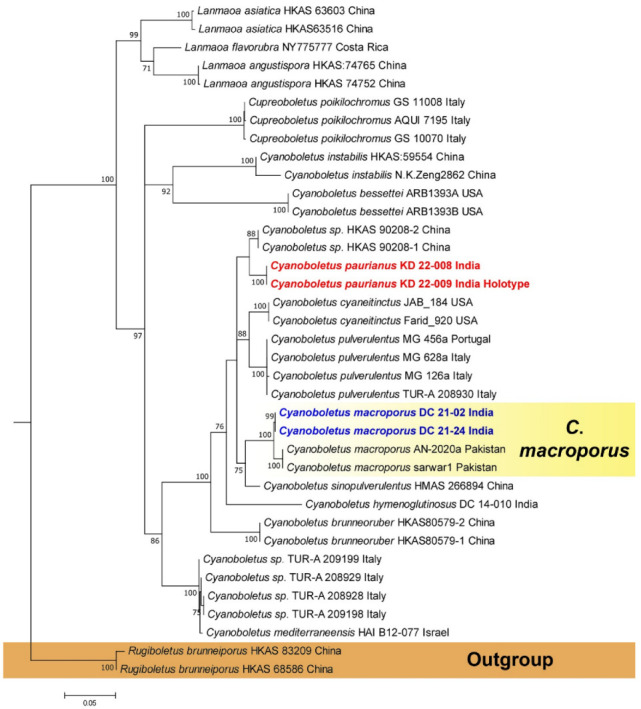
Phylogram generated by Maximum Likelihood analysis based on combined sequence data of nrITS, nrLSU, and *RPB*2 for *Cyanoboletus paurianus*, *C. macroporus,* and allied species. Maximum likelihood bootstrap support values (MLbs) ≥ 70% are shown above or below the branches at nodes. *Cyanoboletus paurianus* and *C. macroporus* are placed in bold red and blue font, respectively, to highlight their phylogenetic positions in the tree.

**Figure 3 jof-09-00754-f003:**
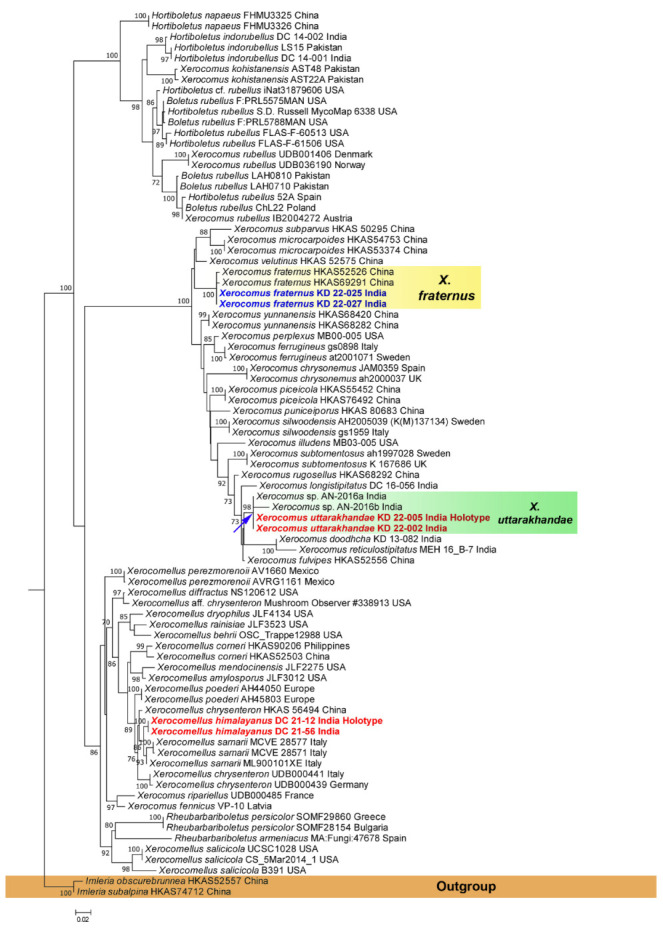
Phylogram generated by Maximum Likelihood analysis based on combined sequence data of nrITS and nrLSU for *Xerocomellus himalayanus*, *Xerocomus uttarakhandae*, *X. fraternus,* and allied species. Maximum likelihood bootstrap support values (MLbs) ≥ 70% are shown above or below the branches at nodes. *Xerocomellus himalayanus* and *Xerocomus uttarakhandae*, *X. fraternus,* are placed in bold blue and red font, respectively, to highlight their phylogenetic positions in the tree.

**Figure 4 jof-09-00754-f004:**
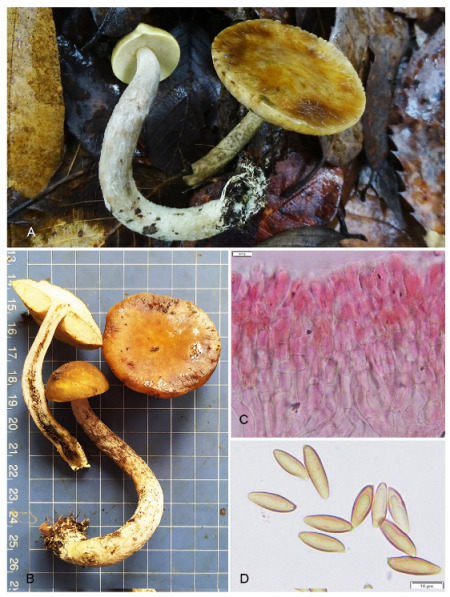
Photographic illustrations ***Leccinellum binderi*** sp. nov. (KD 22-007, holotype). (**A**,**B**) Fresh basidiomata in field and basecamp. (**C**) Elements of pileipellis. (**D**) Basidiospores. Scale bars: (**C**,**D**) = 10 μm.

**Figure 5 jof-09-00754-f005:**
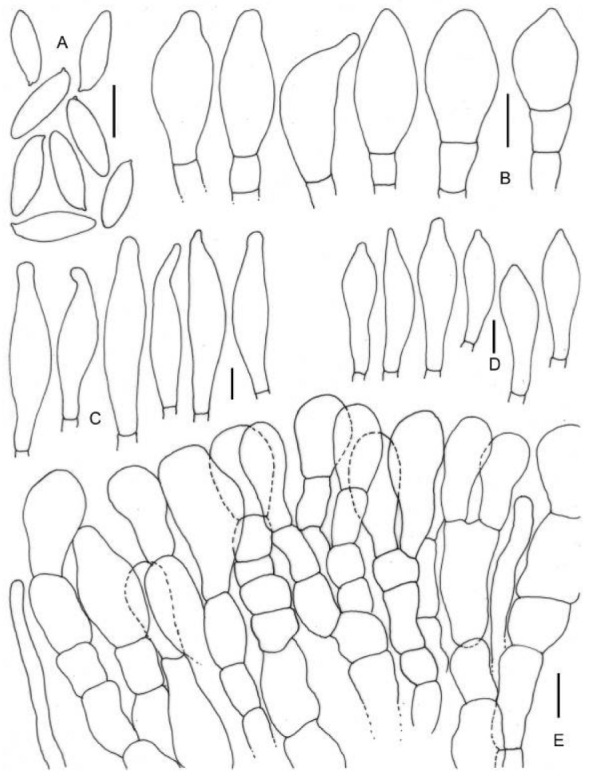
Micromorphological drawings of ***Leccinellum binderi*** sp. nov. (KD 22-007, holotype). (**A**) Basidiospores. (**B**) Caulocystidia. (**C**) Pleurocystidia. (**D**) Cheilocystidia. (**E**) Elements of pileipellis. Scale bars: (**A**–**E**) = 10 μm.

**Figure 6 jof-09-00754-f006:**
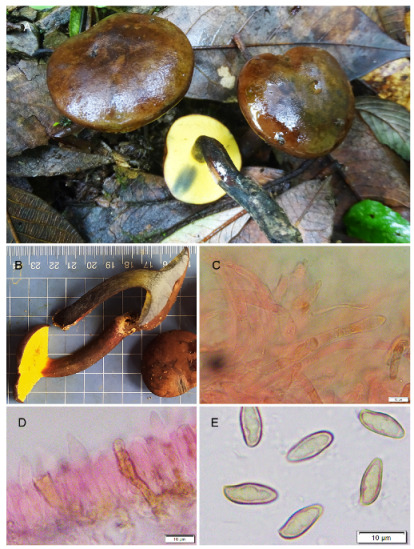
Photographic illustrations ***Cyanoboletus paurianus*** sp. nov. (KD 22-009, holotype). (**A**,**B**) Fresh basidiomata in field and basecamp (**C**) Elements of pileipellis (**D**) Pleurocystidia (**E**) Basidiospores. Scale bar: (**C**,**D**) = 10 μm.

**Figure 7 jof-09-00754-f007:**
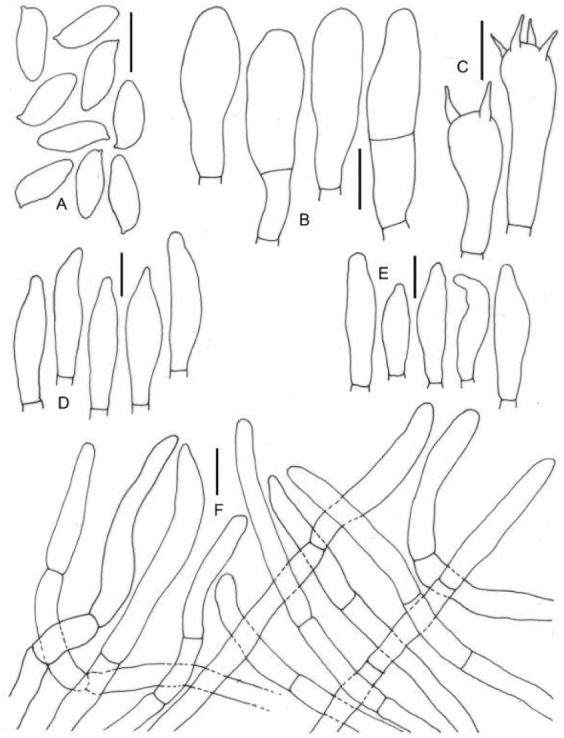
Micromorphological drawings of ***Cyanoboletus paurianus*** sp. nov. (KD 22-009, holotype). (**A**) Basidiospores. (**B**) Caulocystidia. (**C**) Caulobasidia. (**D**) Pleurocystidia. (**E**) Cheilocystidia. (**F**) Elements of pileipellis. Scale bar: (**A**–**F**) =10 μm.

**Figure 8 jof-09-00754-f008:**
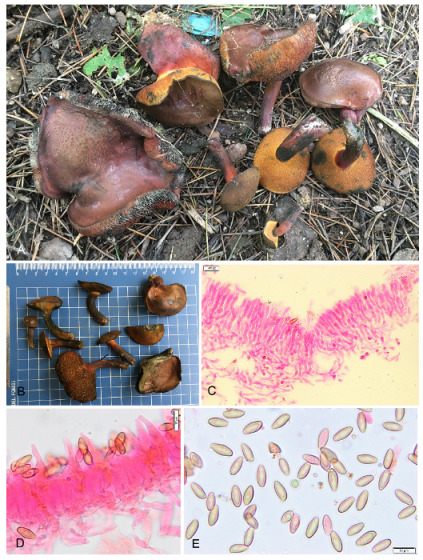
Photographic illustrations of ***Cyanoboletus macroporus*** (DC 21-02). (**A**,**B**) Fresh basidiomata in field and basecamp. (**C**) Elements of pileipellis. (**D**) Hymenial cystidia. (**E**) Basidiospores. Scale bars: C = 20 μm; (**D**,**E**) = 10 μm.

**Figure 9 jof-09-00754-f009:**
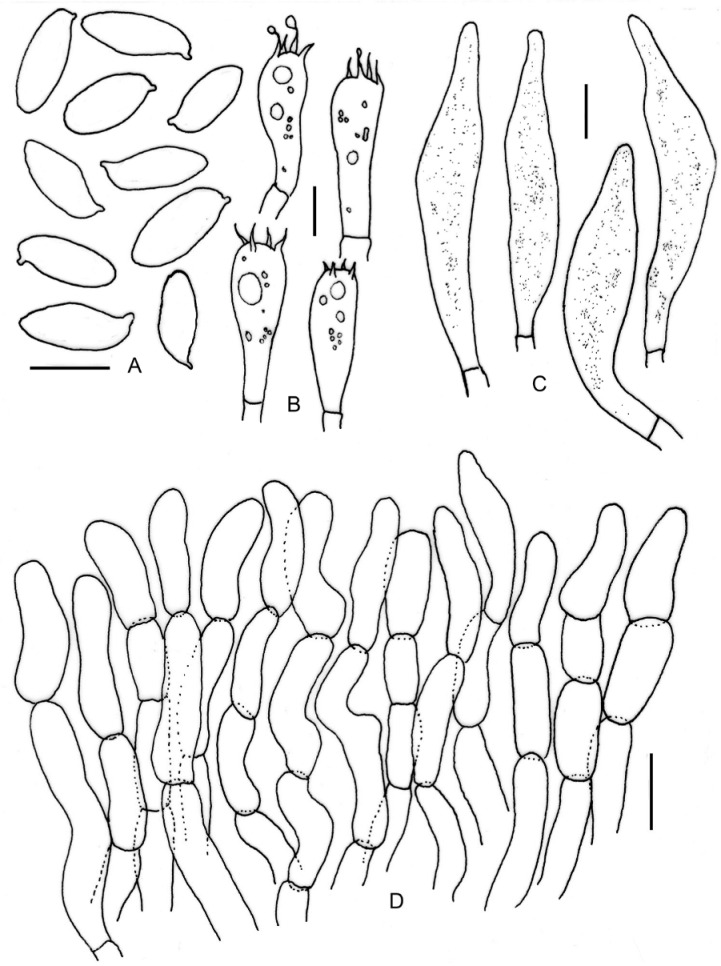
Micromorphological drawings of ***Cyanoboletus macroporus*** (DC 21-02). (**A**) Basidiospores. (**B**) Basidia. (**C**) Hymenial cystidia. (**D**) Elements of pileipellis. Scale bar: (**A**–**D**) = 10 μm.

**Figure 10 jof-09-00754-f010:**
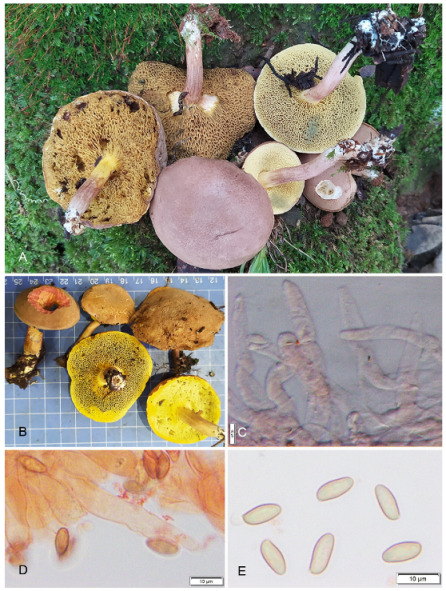
Photographic illustrations of ***Xerocomus uttarakhandae*** sp. nov. (KD 22-005, holotype). (**A**,**B**) Fresh basidiomata in field and basecamp. (**C**) Elements of pileipellis. (**D**) Hymenial cystidia. (**E**) Basidiospores. Scale bars: (**C**–**E**) = 10 μm.

**Figure 11 jof-09-00754-f011:**
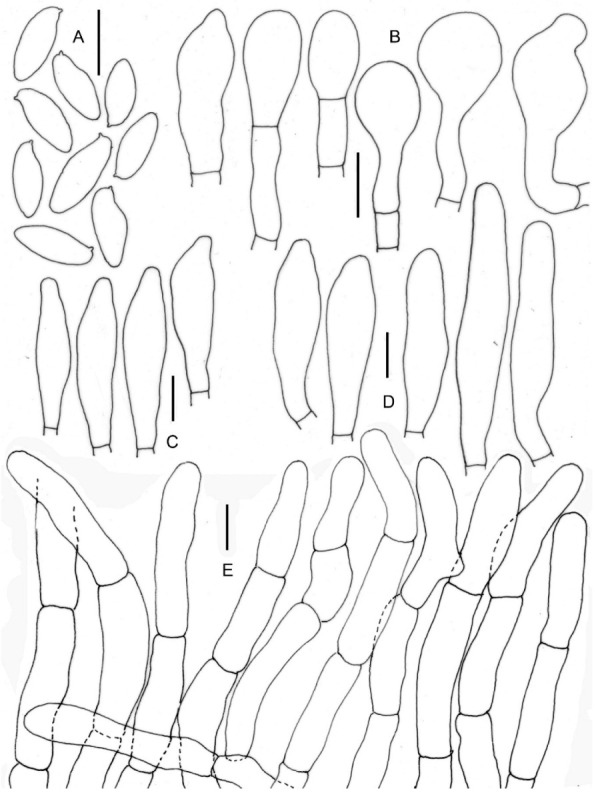
Micromorphological drawings of ***Xerocomus uttarakhandae*** sp. nov. (KD 22-005, holotype). (**A**) Basidiospores. (**B**) Caulocystidia. (**C**) Cheilocystidia. (**D**) Pleurocystidia. (**E**) Elements of pileipellis. Scale bars: (**A**–**E**) =10 μm.

**Figure 12 jof-09-00754-f012:**
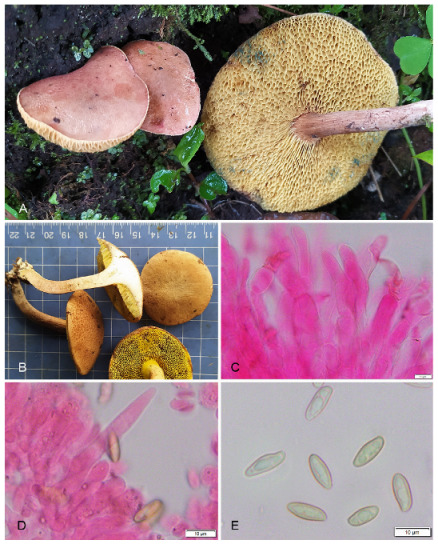
Photographic illustrations of ***Xerocomus fraternus*** (KD 22-025). (**A**,**B**) Fresh basidiomata in field and basecamp. (**C**) Elements of pileipellis. (**D**) Hymenial cystidia. (**E**) Basidiospores. Scale bar: (**C**–**E**) = 10 μm.

**Figure 13 jof-09-00754-f013:**
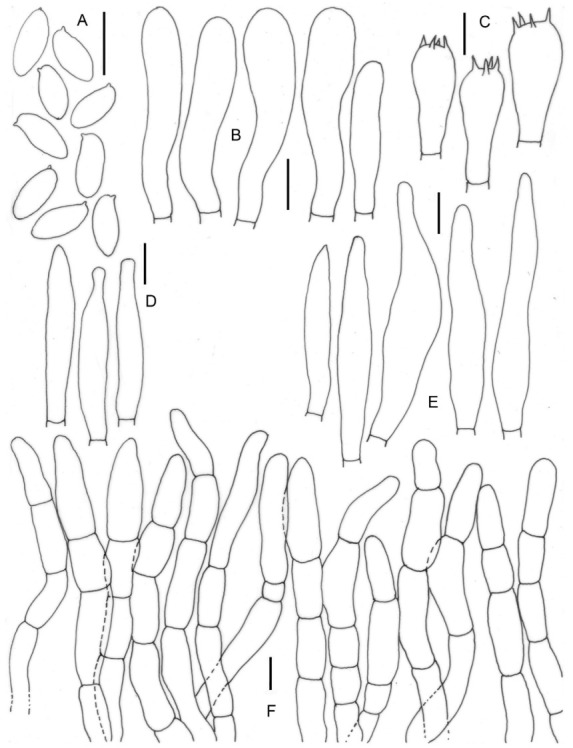
Micromorphological drawings of ***Xerocomus fraternus*** (KD 22-025). (**A**) Basidiospores. (**B**) Caulocystidia. (**C**) Basidia. (**D**) Cheilocystidia. (**E**) Pleurocystidia. (**F**) Elements of pileipellis. Scale bar: (**A**–**F**) = 10 μm.

**Figure 14 jof-09-00754-f014:**
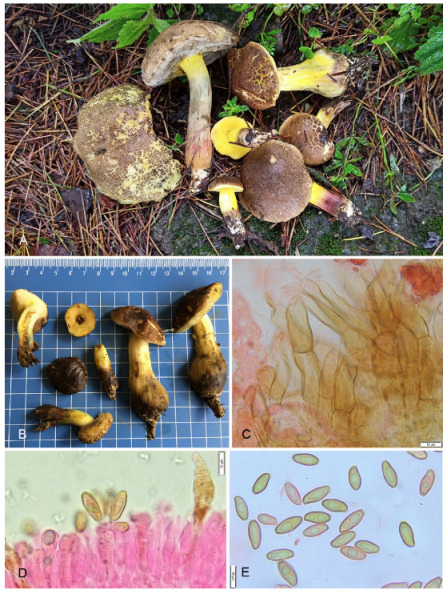
Photographic illustrations of ***Xerocomellus himalayanus*** sp. nov. (DC 21-12). (**A**,**B**) Fresh basidiomata in field and basecamp. (**C**) Elements of pileipellis. (**D**) Hymenial cystidia. (**E**) Basidiospores. Scale bar: (**C**–**E**) = 10 μm.

**Figure 15 jof-09-00754-f015:**
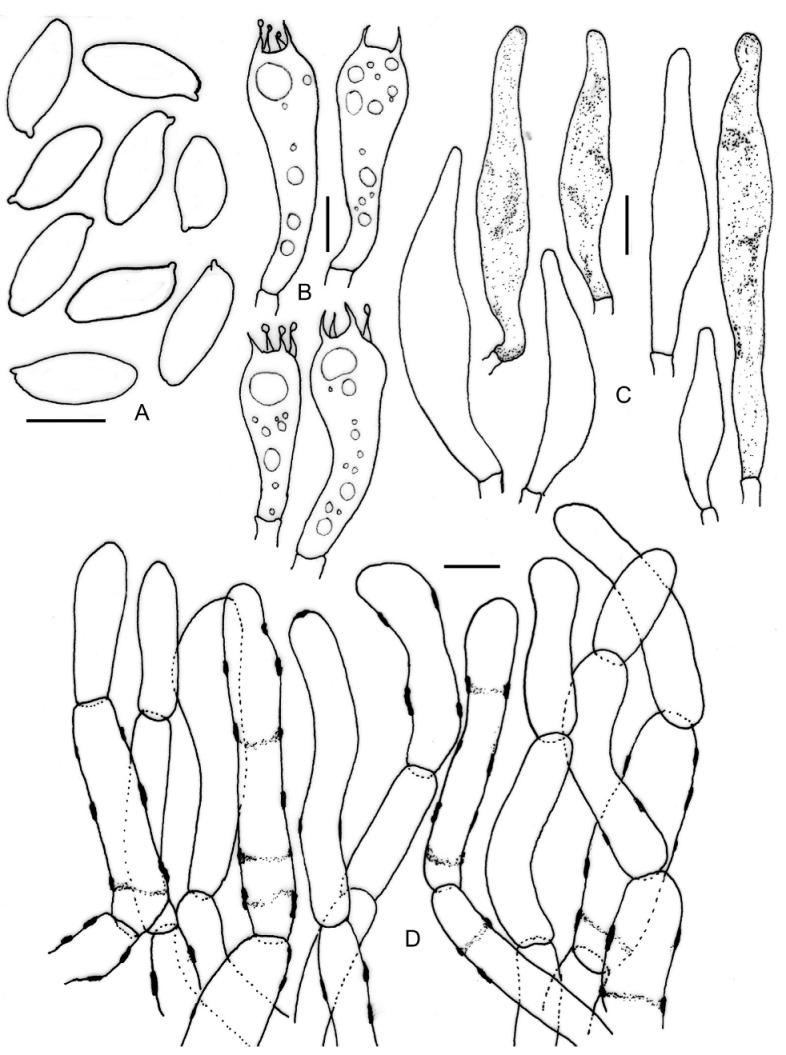
Micromorphological drawings of ***Xerocomellus himalayanus*** sp. nov. (DC 21-12). (**A**) Basidiospores. (**B**) Basidia. (**C**) Hymenial cystidia. (**D**) Elements of pileipellis. Scale bar: (**A**–**D**) = 10 μm.

**Figure 16 jof-09-00754-f016:**
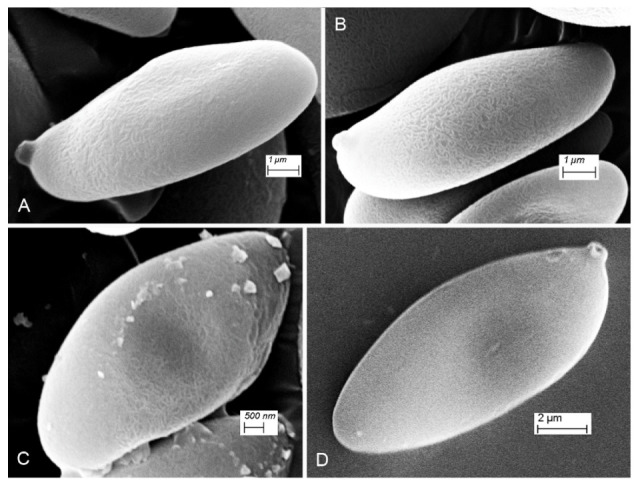
SEM images of basidiospores. (**A**,**B**) Xerocomus uttarakhandae. (**C**) Xerocomus fraternus. (**D**) Xerocomellus himalayanus.

**Table 1 jof-09-00754-t001:** *Leccinellum* and allied sequences used in ML analyses of this study. Newly sequenced collections are in bold.

Species Name (as Reported in GenBank)	Voucher No.	GenBank Accession No.		
		nrLSU	*rpb*2	*tef*1-α
*Borofutus dhakanus*	HKAS73789	JQ928616	JQ928597	JQ928576
*Leccinellum albellum*	MICH KUO-07241101	MK601746	MK766308	MK721100
*Leccinellum alborufescens*	FHMU1908	MK816322	MK816333	MK816330
*Leccinellum alborufescens*	FHMU1758	MK816321	MK816332	MK816329
** *Leccinellum binderi* **	**KD 22-015**	**OQ858379**	**OQ914387**	**OR102316**
** *Leccinellum binderi* **	**KD 22-007**	**OQ858380**	**OQ914386**	**OR102315**
*Leccinellum corsicum*	Buf 4507	KF030347	KF030389	KF030435
*Leccinellum crocipodium*	MICH KUO-07050707	MK601749	MK766311	MK721103
*Leccinellum fujianense*	FHMU2219	MK816319	MK816334	MK816327
*Leccinellum fujianense*	FHMU2223	MK816320	MK816336	MK816328
*Leccinellum indoaurantiacum*	DC 14-019	KT860059	—	—
*Leccinellum lepidum*	K(M)-142974	MK601751	MK766312	MK721105
*Leccinellum pseudoscabrum*	CFMR:DPL-11432	MK601752	MK766313	MK721106
*Leccinellum* sp.	OR0082	—	MZ824749	MZ803024
*Leccinum* aff. *griseum*	KPM-NC-0017381	JN378508	—	JN378449
*Leccinum* aff. *scabrum*	HKAS 57266	KF112442	KF112722	KF112248
*Leccinum album*	Li1072	MW413907	—	MW439267
*Leccinum aurantiacum*	L:0342207	MK601759	MK766318	MK721113
*Leccinum cerinum*	MK11800	AF139692	—	—
*Leccinum duriusculum*	GL4676	AF139699	—	—
*Leccinum duriusculum*	Yang5971	MZ675541	MZ707779	MZ707785
*Leccinum flavostipitatum*	MENMB10801	MH620342	—	—
*Leccinum holopus*	Yang5972	MW413906	MW439258	MW439266
*Leccinum holopus*	9109303	AF139700	—	—
*Leccinum holopus*	MICH: KUO-09150707	MK601763	MK766322	MK721117
*Leccinum manzanitae*	NY-14041 REH-6717	MK601765	MK766324	MK721119
*Leccinum monticola*	HKAS:76669	KF112443	KF112723	KF112249
*Leccinum monticola*	NY-00815448 REH-8591	MK601767	MK766326	MK721121
*Leccinum monticola*	NY-760388 REH-8288	MK601766	MK766325	MK721120
*Leccinum palustre*	MK11107	AF139701	—	—
*Leccinum parascabrum*	Li1700	MW413912	MW439265	MW439272
*Leccinum parascabrum*	Wu1784	MW413911	MW439264	MW439271
*Leccinum pseudoborneense*	WGS965	—	MW439263	—
*Leccinum pseudoborneense*	WGS960	—	MW439262	—
*Leccinum pseudoborneense*	WGS947	MW413908	MW439261	MW439268
*Leccinum rugosiceps*	CFMR BOS-866	MK601770	MK766329	MK721124
*Leccinum scabrum*	HKAS56371	KT990587	KT990423	KT990782
*Leccinum scabrum*	KPM-NC-0017840	JN378515	—	JN378455
*Leccinum variicolor*	Lvar1	AF139706	—	—
*Leccinum versipelle*	FB27	MZ675546	MZ707782	MZ707790
*Leccinum versipelle*	LJW418	MZ675545	MZ707781	MZ707789
*Leccinum versipelle*	CFMR DLC2002-122	MK601778	MK766336	MK721132
*Octaviania japonimontana*	KPM-NC-0017812	JN378486	—	JN378428
*Octaviania tasmanica*	NY-02449788 REH-10066	MK601798	MK766355	MK721152
*Rossbeevera bispora*	GDGM 45639	MK036347	MK350309	—
*Rossbeevera eucyanea*	KPM-NC0023895	KP222896	—	KP222915
*Rossbeevera griseobrunnea*	GDGM45913	MH537793	—	—
*Rossbeevera griseovelutina*	TNS-F-36991	KC552032	—	KC552077
*Rossbeevera vittatispora*	MEL2321058	KP222895	—	KP222911
*Rossbeevera westraliensis*	OSC61480	JN378505	—	JN378445
*Spongiforma thailandica*	BBH:DED 7873	NG_042464	—	—

**Table 2 jof-09-00754-t002:** *Cyanoboletus* and allied sequences used in ML analyses of this study. Newly sequenced collections are in bold.

Species Name (as Reported in GenBank)	Voucher No.	GenBank Accession No.		
		nrITS	nrLSU	*rpb*2
*Cupreoboletus poikilochromus*	GS 10070	KT157051	KT157060	KT157068
*Cupreoboletus poikilochromus*	GS 11008	KT157050	KT157059	KT157067
*Cupreoboletus poikilochromus*	AQUI 7195	KT157052	KT157061	—
*Cyanoboletus bessettei*	ARB1393B	MW675738	—	MW737458
*Cyanoboletus bessettei*	ARB1393A	MW675737	MW662571	MW737457
*Cyanoboletus brunneoruber*	HKAS80579-2	—	KT990569	KT990402
*Cyanoboletus brunneoruber*	HKAS80579-1	—	KT990568	KT990401
*Cyanoboletus cyaneitinctus*	JAB_184	MW675731	MW662584	MW737467
*Cyanoboletus cyaneitinctus*	Farid_920	MW675744	MW662579	MW737465
*Cyanoboletus hymenoglutinosus*	DC 14-010	KT907355	KT860060	—
*Cyanoboletus instabilis*	N.K.Zeng2862	MG030473	MG030466	—
*Cyanoboletus instabilis*	HKAS:59554	—	KF112412	KF112698
*Cyanoboletus macroporus*	sarwar1	MW369503	—	—
*Cyanoboletus macroporus*	AN-2020a	MW045557	—	—
** *Cyanoboletus macroporus* **	**DC 21-02**	**OQ860238**	**OQ860239**	**ON364552**
** *Cyanoboletus macroporus* **	**DC 21-24**	**OQ860240**	**OQ860241**	**OQ876894**
*Cyanoboletus mediterraneensis*	HAI B12-077	OM801199	OM801212	—
** *Cyanoboletus paurianus* **	**KD 22-008**	—	**OQ859920**	**OQ914389**
** *Cyanoboletus paurianus* **	**KD 22-009**	—	**OQ859919**	**OQ914388**
*Cyanoboletus pulverulentus*	MG 126a	KT157053	KT157062	—
*Cyanoboletus pulverulentus*	MG 456a	KT157054	KT157063	—
*Cyanoboletus pulverulentus*	MG 628a	KT157055	KT157064	KT157069
*Cyanoboletus pulverulentus*	TUR-A 208930	MZ265186	—	MZ265200
*Cyanoboletus sinopulvirulentus*	HMAS266894	KC579402	—	—
*Cyanoboletus* sp.	HKAS90208-2	—	—	KT990405
*Cyanoboletus* sp.	HKAS90208-1	—	KT990571	KT990404
*Cyanoboletus* sp.	TUR-A 209199	MZ265183	MZ265198	—
*Cyanoboletus* sp.	TUR-A 208928	MZ265179	MZ265194	—
*Cyanoboletus* sp.	TUR-A 209198	MZ265182	MZ265197	—
*Cyanoboletus* sp.	TUR-A 208929	MZ265181	MZ265196	—
*Lanmaoa angustispora*	HKAS:74765	—	KF112322	KF112680
*Lanmaoa angustispora*	HKAS 74752	—	KM605139	KM605177
*Lanmaoa asiatica*	HKAS63516	—	KT990584	KT990419
*Lanmaoa asiatica*	HKAS 63603	—	KM605143	KM605176
*Lanmaoa flavorubra*	NY775777	—	JQ924339	KF112681
*Rugiboletus brunneiporus*	HKAS 83209	—	KM605134	KM605168
*Rugiboletus brunneiporus*	HKAS 68586	—	KF112402	KF112719

**Table 3 jof-09-00754-t003:** *Xerocomus* and allied sequences used in ML analyses of this study. Newly sequenced collections are in bold.

Species Name (as Reported in GenBank)	Voucher No.	GenBank Accession No.	
		nrITS	nrLSU
*Boletus rubellus*	F:PRL5575MAN	GQ166888	—
*Boletus rubellus*	F:PRL5788MAN	GQ166883	—
*Boletus rubellus*	ChL22	KX438318	—
*Boletus rubellus*	LAH0710	KJ802928	—
*Boletus rubellus*	LAH0810	KJ802929	—
*Hortiboletus* cf. *rubellus*	iNat31879606	MN498119	—
*Hortiboletus indorubellus*	DC 14-002	KT319647	—
*Hortiboletus indorubellus*	DC 14-001	—	KU566807
*Hortiboletus indorubellus*	LS15	MK002767	MK002872
*Hortiboletus kohistanensis*	AST48	MG988192	MG988187
*Hortiboletus kohistanensis*	AST22A	MG988193	—
*Hortiboletus napaeus*	FHMU3325	MT646445	MT646438
*Hortiboletus napaeus*	FHMU3326	MT646440	MT646433
*Hortiboletus rubellus*	FLAS-F-61506	MH211937	—
*Hortiboletus rubellus*	FLAS-F-60513	MH211664	—
*Hortiboletus rubellus*	52A	MN652008	—
*Hortiboletus rubellus*	S.D. Russell MycoMap 6338	MK560106	—
*Imleria obscurebrunnea*	HKAS52557	KC215207	KC215220
*Imleria subalpina*	HKAS74712	KC215208	KC215218
*Rheubarbariboletus armeniacus*	MA:Fungi:47678	AJ419221	—
*Rheubarbariboletus persicolor*	SOMF28154	MH011932	—
*Rheubarbariboletus persicolor*	SOMF29860	MH011931	—
*Xerocomellus salicicola*	UCSC1028	KU144793	KU144794
*Xerocomellus* aff. *chrysenteron*	Mushroom Observer #338913	ON705310	—
*Xerocomellus amylosporus*	JLF3012	KM213635	KU144742
*Xerocomellus behrii*	OSC_Trappe12988	KJ882288	—
*Xerocomellus chrysenteron*	HKAS:56494		KF112357
*Xerocomellus chrysenteron*	—	UDB000441	—
*Xerocomellus chrysenteron*	—	UDB000439	—
*Xerocomellus corneri*	HKAS90206	—	KT990669
*Xerocomellus corneri*	HKAS52503	—	KT990668
*Xerocomellus diffractus*	NS120612	KM213650	KM213651
*Xerocomellus dryophilus*	JLF4134	KX534076	KY659593
** *Xerocomellus himalayanus* **	**DC 21-56**	**OQ847959**	**OQ847979**
** *Xerocomellus himalayanus* **	**DC 21-12**	**OQ847832**	**OQ847962**
*Xerocomellus mendocinensis*	JLF2275	KM213653	KM213654
*Xerocomellus perezmorenoi*	AV1660	OK350679	OK350681
*Xerocomellus perezmorenoi*	AVRG1161	OK350680	OK350682
*Xerocomellus poederi*	AH44050	KU355475	KU355488
*Xerocomellus poederi*	AH45803	KU355480	KU355491
*Xerocomellus rainisiae*	JLF3523	KU144789	KU144790
*Xerocomellus salicicola*	B391	MW675727	MW662569
*Xerocomellus salicicola*	CS_5Mar2014_1	KU144791	KU144792
*Xerocomellus sarnarii*	ML900101XE	MH011930	—
*Xerocomellus sarnarii*	MCVE 28571	KT271745	—
*Xerocomellus sarnarii*	MCVE 28577	KT271749	—
*Xerocomus chrysonemus*	ah2000037	DQ066381	—
*Xerocomus chrysonemus*	JAM0359	—	KF040544
*Xerocomus doodhcha*	KD 13-082	KR611867	KU566806
*Xerocomus fennicus*	VP-10	KT692929	—
*Xerocomus ferrugineus*	gs0898	DQ066403	—
*Xerocomus ferrugineus*	at2001071	DQ066402	—
** *Xerocomus fraternus* **	**KD 22-025**	**OQ776920**	**OQ771932**
** *Xerocomus fraternus* **	**KD 22-027**	**OQ776919**	**OQ771933**
*Xerocomus fraternus*	HKAS52526	—	KT990682
*Xerocomus fraternus*	HKAS69291	—	KT990683
*Xerocomus fulvipes*	HKAS52556	—	KT990672
*Xerocomus illudens*	MB03-005	JQ003658	—
*Xerocomus longistipitatus*	DC 16-056	KY008398	—
*Xerocomus microcarpoides*	HKAS54753	—	KT990680
*Xerocomus microcarpoides*	HKAS53374	—	KT990679
*Xerocomus perplexus*	MB00-005	JQ003657	JQ003702
*Xerocomus piceicola*	HKAS55452	—	KT990685
*Xerocomus piceicola*	HKAS76492	—	KT990684
*Xerocomus puniceiporus*	HKAS 80683	—	KU974141
*Xerocomus reticulostipitatus*	MEH 16_B-7	MF167353	—
*Xerocomus ripariellus*	—	UDB000485	
*Xerocomus rubellus*	—	UDB036190	—
*Xerocomus rubellus*	—	UDB001406	—
*Xerocomus rubellus*	IB2004272	EF644119	—
*Xerocomus rugosellus*	HKAS68292	—	KT990686
*Xerocomus silwoodensis*	AH2005039 (K(M)137134)	DQ438143	—
*Xerocomus silwoodensis*	gs1959	DQ066375	—
*Xerocomus* sp.	AN-2016a	KU761593	—
*Xerocomus* sp.	AN-2016b	KU761592	—
*Xerocomus subparvus*	HKAS50295	—	KT990667
*Xerocomus subtomentosus*	ah1997028	DQ066370	—
*Xerocomus subtomentosus*	K 167686	JQ967281	JQ967238
** *Xerocomus uttarakhandae* **	**KD 22-002**	**OQ748035**	**OQ748038**
** *Xerocomus uttarakhandae* **	**KD 22-005**	**OQ748036**	**OQ748037**
*Xerocomus velutinus*	HKAS 52575	—	KF112393
*Xerocomus yunnanensis*	HKAS68420	—	KT990690
*Xerocomus yunnanensis*	HKAS68282	—	KT990691

## Data Availability

The sequences presented in this study are openly available in https://www.ncbi.nlm.nih.gov/, accessed on 21 June 2023. All new taxa were registered in MycoBank (http://www.mycobank.org/, accessed on 21 June 2023).

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
