# Peer review of "Four Novel Species and Two New Records of Boletes from India"

_jof, 2023, doi:10.3390/jof9070754_

Round 1

Reviewer 1 Report

This MS presennted well. Only few issues-

1. Mycobank should be MycoBank

2. Make sure LSU or nrLSU throughout the text.

3. Species epithet name: Use species characters of a species to keep the species name rather than locality names or mycologist name. You may keep one species name in honor of mycologist of locality name but not for all.

4. References 2, 17, 18: Hosen, I. should be Hosen, M.I.

5. 6: Check title words some words should be lower cases and as well as other titles.

6. Check reference abbreviate form or elaborate form/follow author instructions of journal.

7. Would you mind to add Kaziboletus and Spongispora in your phylogenetic tree and see the placement of them?

Author Response

Dear reviewer 1,

Thank you very much for your valuable comments. Please see my point-to-point response in the attachment.

Best regards,

Komsit Wisitrassameewong

Reviewer 2 Report

Referee's comments

 Manuscript title: Boletes from India: Four new species and two new records

 In the manuscript were discovered four new species and two new records from boletes fungi based on multi-locus phylogeny and morphological data.

In general, this is a well-written manuscript

The topic of the work sounds interesting, but I have suggestions and noticed a few issues in the manuscript (listed below), that need to be addressed before acceptance.

Minor edits:

Manuscript title is clear but alternative title “Four novel species and two new records from Boletes Fungi of India”

The abstract is overall well written. But too many common words in abstract and the same categories of characters are listed for each new species of Boletes, which do not really tell the distinguished characters of the new species from other species and each other in the genus/genera, so please provide these information

The introduction and methods sections were all sufficient and clear. But I will recommend write some information about how many species Boletes are occurred and who is studied before you in India in part of Introduction

Results section were all sufficient and clear but please remove Key to the studied genera to Discussion part of article.

Discussion

1. I recommend you write currently how many Bolete species occur in India with your novel species and new records in the Discussion part.

2. I suggest you also write some discussion of ecological information of your new species and other Bolete species in this part 

Here better provide another scientific conclusion about Bolete species from your phylogenetic and morphological outputs with combined previously published results 

Suggestion for Table

Table 1,2,3. If possible please provide substrates and collected location of the used species in table such as which country and host plants/habitat   

Some English language editorial changes are needed in the text

Author Response

Dear reviewer 2,

Thank you very much for your valuable comments. Please see my point-to-point response in the attachment.

Best regards,

Komsit
